# Sustainability at Play: Educational Design Research for the Development of a Digital Educational Resource for Primary Education

**Patrícia Sá** [1,*], **Patrícia Christine Silva** [1], **Joana Peixinho** [2], **Ana Figueiras** [3] **and Ana V. Rodrigues** [1]

[1] Research Centre on Didactics and Technology in the Education of Trainers, Department of Education and Psychology, University of Aveiro, 3810-193 Aveiro, Portugal
[2] Intermunicipal Community Viseu Dão Lafões, 3460-613 Tondela, Portugal
[3] Centre for Research in Applied Communication, Culture, and New Technologies, Universidade Lusófona, Campo Grande, 1749-024 Lisboa, Portugal
[*] Correspondence: patriciasa@ua.pt

**Abstract:** Quality education is an indispensable element for the successful implementation of the 2030 Agenda for Sustainable Development, as it equips all children with the essential skills to promote sustainable development within the context of their lifestyles, in line with the fourth Sustainable Development Goal (SDG4). This will have to be an innovative education, not only in the contents and guidelines to be followed but also in the educational strategies and resources to be used. This article aims to present and describe the methodology used to develop the digital educational resource (DER) "Sustainability at Play," a serious game intended for primary education and based on the concept of the ecological footprint. This DER was developed through educational design research (EDR) by a multidisciplinary team within a project to create Digital Educational Resources for Primary Education. The EDR approach was organized in four phases—Phase I—Problem Analysis, Phase II—Exploration of Possible Solutions, Phase III—Design, and Phase IV—Reflection—allowing for an iterative DER development process shared by different participants (researchers, illustrators, web designers, primary school teachers and students). As a result, this study enabled not only the development of an innovative DER to work on issues related to education for sustainability in primary school but also the understanding and validation of the suitability of the EDR methodology for the development of this type of educational resource.

**Keywords:** educational design research; education for sustainability; digital educational resource; primary education

## 1. Introduction

The central role that education assumes in implementing the 2030 Agenda for Sustainable Development is recognized and reinforced in several international documents and guidelines (Leicht et al. 2018; UNESCO 2015, 2017a). As a result, quality education explicitly emerges as one of the Sustainable Development Goals (SDG4). However, in the 2030 Agenda document, education is not limited to SDG4. It appears in several other SDGs and is also recognized as an essential means of achieving this agenda (Leicht et al. 2018; UNESCO 2015, 2017b).

According to UNESCO, one of the main objectives of education for sustainability (EduS) is to empower citizens to make informed decisions and act responsibly toward environmental integrity, economic viability, and social justice (UNESCO 2017a). Implementing a cross-cutting and integrated EduS orientation can include more than just new content in the curriculum. It requires a transformative, action-oriented pedagogy focused on learning from the earliest years of schooling, which makes subjects responsible for exercising conscious citizenship and solidarity. For the innovation and reorientation that this

implementation requires, research in education, teacher education, and the development of new educational resources will be essential. This article focuses mainly on educational design research (EDR) and its contribution to developing new educational resources and innovation in education in the field of EduS.

According to Napal et al. (2020), one of the fields that most urgently needs to utilize the potential benefits of digital technologies to transform learning is sustainability, evidencing the pertinence of studies such as the one presented here. In particular, several authors emphasize the relevance of serious games in an EduS approach (Neset et al. 2020; Ouariachi et al. 2019; Stanitsas et al. 2019). Serious games are important since they allow players to experience unfamiliar circumstances that are not possible in real life, enabling awareness to be raised and changes in attitude and behavior in players to be promoted (Ouariachi et al. 2019; Stanitsas et al. 2019). Neset et al. (2020) emphasized the importance of these resources in the teaching and learning process, stating that serious games can support teachers to strengthen their EduS, as they provide an experience of climate adaptation based on systems thinking and action orientation.

The growing acknowledgment of EduS and the perceived benefit of serious games for teaching students about sustainability are both recognized arguments that highlight the relevance of this field of study (Hallinger et al. 2020).

In recent years, there has been a growing trend towards creating Digital Educational Resources (DER) for early years of schooling in the field of EduS. Digital games have emerged as a popular didactic strategy, providing students a playful and meaningful learning experience (Veronica and Calvano 2020; Chappin et al. 2017; Jesus et al. 2021; Leal et al. 2022; Oliveira et al. 2021; Sá et al. 2013; Souza et al. 2020; Vestena and Bem 2020). The following studies showcase digital games with educational purposes underlying sustainability themes. These studies provide detailed descriptions of the games, outline the validation process involving the target audience, and present the outcomes observed during the implementation.

The prototype of the game "BioSolo" (Leal et al. 2022), subordinated to the theme of environmental sustainability, was developed for children aged 6 to 11 years old and aims to raise awareness about soil fauna. With the soil as the main screen, the player is challenged to uncover various organisms found in it and classify them appropriately. Points are accumulated as these organisms are classified correctly and may be doubled by taking the final quiz related to the theme explored. The prototype was evaluated through interviews with experts from various areas, and the results point to it being an adequate resource for its proposed purposes.

The game "RECICLAPPSM" (Vestena and Bem 2020), developed for primary school children, aims to raise awareness about the proper treatment of the solid waste we produce, considering its toxicity and hazardousness. The player is challenged through drag-and-drop dynamics to place the waste in the correct containers, gaining points for its correct separation. Separating solid waste requires gloves that are suitable for this purpose. If the player does not use them correctly, they lose points. The game is completed after the six levels have been reached. It was tested with a group of children, and the results of this implementation reveal the game's suitability as a contribution to developing the learning objectives it proposes.

The game "SimSustentabilidade" (Jesus et al. 2021) is configured as a simulation and strategy game, challenging the player to build a city (companies, research centers) and manage financial and environmental resources. The game develops with the player's decision-making to manage the city sustainably, creating the least environmental impact possible. They must guarantee economic growth and eliminate polluting gas emissions by planting forests. The evaluation carried out on the game indicated that the game could be a potentiator of more learning about sustainability in a ludic way, but limitations include the small, minimalist interface and the few construction options offered to the player.

The game "Produtos Perigosos ou Sustentáveis" (Oliveira et al. 2021) aims to develop skills related to ecotoxicology, specifically environmental contamination by chemicals. It

was developed for children attending the 9th grade. The average duration of the game is about 50 min, and it was developed to be used in a classroom context. It consists of 13 stages, and in each of the game's situations, the player can win up to 10 points, for a maximum score of 130 points. In a game situation, the player completes the spaces with letters (forming words) to fill the sentence with the missing words. Points are removed if the player needs hints to progress in the game or for any mistakes. The 10 points are received in each phase if the player manages to advance without a request for hints and without making a mistake. By conducting a pre- and post-test and questionnaire, it was determined that the children learned, and the evaluation carried out of its functionality, usability, and playful component was positive.

Veronica and Calvano (2020) developed a game entitled "SeAdventure" targeting children in the 4th grade, with a central theme of ocean literacy. As a complement, an introductory and contextualizing video was developed that focuses on the issue of marine debris, titled "A plastic ocean." The game, through an avatar, takes players on an underwater journey, during which they encounter a variety of garbage. Four endangered species, including tuna, shark, turtle, and seahorse, are featured as characters. The objective is for children to guide their characters to find food without consuming the surrounding garbage, raising awareness about the challenges these species face in survival. The additional information provided also promotes knowledge about the species' habitat, lifestyle, food, and other perils they face in the ocean. Players accumulate points by feeding their characters correctly but lose points if they consume solid waste. Evaluations carried out led to the determination that the game is an enabler of further learning related to ocean literacy.

The game "Universal Machine Ecological: U.M.E." (Souza et al. 2020) entails a game developed between the 2nd and 5th grades that addresses the issue of some types of pollution. A droid named U.M.E. is the character of the game, who aims to raise awareness about the issue of environmental pollution. At this stage there is no time limit for reading in order to respect the individuality of the children. One of the other goals is to collect the garbage within a time limit of 20 s; if the player cannot meet this challenge they lose. In the last phase, the objective is to raise awareness about the consequences of pollution and the importance of change in this context.

The game "Catan: Oil Springs" (Chappin et al. 2017) is an entertainment game with incorporated sustainability concepts. The idea of this game is for it to act as an addition to the board game "Settlers of Catan" released in October 2011. The game introduces oil as an additional resource within the world of Catan, complementing the regular gameplay. Through the game dynamics, players are confronted with real sustainability issues and are compelled to reevaluate their strategies, not only for their own victory but also to address broader sustainability concerns, particularly related to climate change. Oil in the game serves as a substitute for fossil fuels and other mineral resources, highlighting its benefits while emphasizing the risks of pollution and adverse effects associated with climate change. The scenario aims to raise awareness about pressing sustainability challenges, with the ultimate goal of fostering a transformation in thinking and consciousness, and, as a result, a change in behavior.

Another example of a digital game is the Courseware SeRe® "The Human Being and Natural Resources" (Sá et al. 2013). This courseware emerged from the awareness of the lack of quality digital educational resources for science education towards education for sustainable development (ESD). The objective of this project was to encourage primary school teachers in a new way of understanding science education, thereby promoting the integration of ESD principles into their science lessons. This resource allows for the adoption of new orientations, digital educational resources, and new science teaching and learning strategies.

Although not identified as such by the authors, part of these games could be classified as "serious games." Clark C. Abt (1987) coined the term "serious games" in a book with the same title, arguing that games could be used for more than just entertainment and could have educational, training, and simulation intentions. In his definition, he approaches these

games as being constructed with an explicit and carefully thought out educational purpose and not intended to be played primarily for entertainment. However, the term has since been widely adopted to designate games with a primary purpose other than entertainment, such as education, training, health, and social change.

Because this definition is too broad, Ian Bogost (2010) uses the term "persuasive games," since he believes they can bring attention to issues and stimulate players to think critically about their values and beliefs. By using the term "persuasive games," Bogost emphasizes that these games can be used as a tool for persuasion in the same way as advertising or propaganda, therefore making them suitable to enable players to understand complex issues, challenge their assumptions and biases, and encourage them to take action. In this way, he sees games as a powerful medium for social and political change. According to Michael and Chen (2006), serious games are a tool for promoting learning and are potential promoters of behavioral change, as they can engage players in interactive and immersive experiences that enhance their motivation and interest in the topic. Wouters et al. (2009) also suggested that serious games can be used to promote the mobilization and/or development of different dimensions of various competencies (e.g., knowledge, skills, attitudes, values, dispositions). Furthermore, serious games can positively impact learners' motivation and engagement. Boyle et al. (2012) found that digital entertainment games, including serious games, are associated with high levels of engagement, which is a crucial factor in promoting learning and attitude change.

This article aims to present and describe the methodology used to develop the DER "Sustainability at Play" game (Sá et al. 2021), a serious game aimed at primary education. This DER is part of the project Digital Educational Resources for Primary Education (POCH-04-5267-FSE-000124), coordinated by the Directorate-General for Education (Ministry of Education of Portugal) and of which DER_Sciences is a part. The development of DER_Sciences was founded in the didactic guidelines designed under the Training Program in Experimental Teaching of Sciences for Primary School Teachers (PFEEC) (Despacho No. 2143/2007, 9 February and Despacho No. 701/2009, 9 January) that took place over four school years, between 2006 and 2010, with the purpose of, through the development of professional skills of primary school teachers, increasing the scientific literacy levels of Portuguese students.

DER_Sciences was developed by a multidisciplinary team of researchers from the University of Aveiro (UA) and the New University (UN) of Lisbon. This multidisciplinary team constituted 11 specialists in science teaching and 20 members of the multimedia team responsible for different specialties (e.g., digital creation project management, illustration, visualization and management of website content, design, video game production, programming, sound design, digital communication). The main objectives of this project were (i) to develop and evaluate multimedia resources on science topics for primary school students 6–10 years old that promote autonomous learning and (ii) to develop a website where DER_Sciences was made available with free access, promoting the integration of digital technologies into teaching and learning processes.

Within the dedicated domain for natural sciences on the developed website (https://redge.dge.mec.pt/ilha/ accessed on 29 June 2023), a section containing didactic proposals for exploring sustainability-related issues with children is available. These proposals encompass topics such as human demography and the conservation of the planet's resources.

## 2. Materials and Methods

As previously mentioned, this paper aims to present and describe the methodology used to develop the DER "Sustainability at Play" game. The research question guiding the study was the following: How can a DER for education for sustainability (EduS) in primary education be developed? Given its nature, the characteristics that define it, and its purposes, the EDR methodology emerged as a suitable methodology since it intends to support in research the development of solutions to complex educational problems

through a reasoned, iterative, collaborative process involving a multidisciplinary team for this purpose.

*2.1. Design and Development of the Digital Educational Resource "Sustainability at Play"*

The "Sustainability at Play" game is a digital educational resource developed by a multidisciplinary team based on the EDR approach. EDR is one of the approaches integrated into the design research perspective, which emerged at the end of the 20th century/beginning of the 21st century as an alternative to the methodologies that had been used in the field of educational research (and that were being pointed out by several authors as limiting the potential of educational research). Design research has been referred to in the literature not so much as an approach per se but rather as a set of approaches " . . . with the intent of producing new theories, artefacts, and practices that account for and potentially impact learning and teaching in a naturalistic setting" (Barab and Squire 2004, p. 2). Although different designations can be found for the various DR approaches—e.g., design-based research (Kelly 2003), development research (Van den Akker 1999), formative research (Newman 1990), and educational design research (McKenney and Reeves 2019, 2021; Plomp and Nieveen 2013)—they are very similar in their characteristics, aims, and proposed implementation models. According to McKenney and Reeves (2013, 2021), EDR can be defined as a "genre" of research emphasizing the iterative construction of solutions/proposals to complex educational problems. These solutions can be educational products, processes, programs, or even policies.

Concerning its main characteristics, several authors have described EDR as pragmatic (focused on the production of usable know-how and valuable solutions to complex and real-world problems), evidence-based (based on theoretical frameworks of reference), interventionist (aimed at promoting changes in a particular educational context), iterative (developed through different cycles of design, development, testing, and reformulation), and collaborative (involving multiple partners in multidisciplinary teams) (McKenney and Reeves 2013, 2021).

In addition, with regard to the process, different proposals for the EDR approach can be found, with variations in terms of the terminology used and even the models presented. However, within this diversity, there are some common aspects that several authors have identified in the reference literature. Plomp (2013) systematized the following aspects:

1.  EDR relies on scientific knowledge to support the work developed;
2.  The process of implementing the approach results in the construction/production of scientific knowledge (contributing to a broader understanding of the problem under study/to be addressed);
3.  EDR considers different phases for its implementation. Although the phases considered and their designations present some differences (depending on the author), no matter the considered proposal, EDR entails an analysis/guidance phase, a design/development phase, and an evaluation/retrospective phase. Some authors have proposed models with four phases (e.g., Gravemeijer and Cobb 2006), and others have proposed four-phase EDR organization models (Reeves 2000, 2006). All of these phases can be cyclically repeated throughout the process.

In the present study, the model proposed by Reeves (2000, 2006) and McKenney and Reeves (2021) was adopted, considering four phases for implementing the EDR approach: Phase I—Problem Analysis, Phase II—Exploration of Possible Solutions, Phase III—Design, and Phase IV—Reflection. The following table (Table 1) systematizes the phases followed throughout the EDR development process, explaining the main tasks performed in each phase, as well as the two iterative cycles implemented from problem identification and characterization to the proposed solution—the final version of the resource.

**Table 1.** Phases followed throughout the EDR development process.

| EDR Phases | Tasks | Participants |
|---|---|---|
| Phase I—(February 2019) Problem Analysis (Reeves 2000, 2006; McKenney and Reeves 2019, 2021) | Problem identification and analysis Preliminary research Literature review | Multidisciplinary research team |
| Phase II—Exploration of Possible Solutions | Searching for solutions and design proposals for the development of the intended resource Exploring ideas and defining the structure of the game and the sequence of activities to be included (design skeleton) Setting up the database with the ecological footprint (EP) values needed for the game | Multidisciplinary research team |
| Phase III— Design/Construction of Iterative Testing and Reformulation Cycles (Reeves 2000, 2006; McKenney and Reeves 2019, 2021) | *Cycle 1—Prototype 1*<br>- Prototype 1 Construction Definition of the navigability characteristics Elaboration of the first proposals for the illustrations (creation of prototypes for various game scenarios)<br>- Prototype 1 Validation Definition of strategies, methods, and instruments for the validation Identification of experts and end-users to participate in the intended validation Validation of the contents of the EP database Validation of the proposed illustrations Validation of the navigability Analysis of the results of the validations performed and identification of the changes to be made<br><br>*Cycle 2—Prototype 2*<br>- Prototype 2 Construction Review and reformulation of the game structure and initial prototypes (considering the aspects pointed out/discussed in the validation process and in the meetings of the extended multidisciplinary team) Reformulation of the illustrations of the game's basic scenarios Building the first complete version of the resource (with the new illustrations and the complete scenarios)<br>- Evaluation—Prototype 2 Pilot Definition of methods, strategies, and instruments for data collection and analysis to be used in the pilot study to be carried out Pilot study of the first complete version of the resource with primary school students Analysis of the results of the validations performed and identification of the changes to be made | Researchers (large multidisciplinary team) Web designers Illustrators Specialists Primary school students |
| Phase IV—(February 2021) Reflection (Reeves 2000, 2006; McKenney and Reeves 2019, 2021) | Final evaluation/reflection on the process and the resulting resource | |

### 2.2. Phase I—Problem Analysis

The first phase of the ERD involves defining and analyzing the problem to be tackled. In this case, the research question that guided the whole process was the following: How can digital educational resources (DER) for education for sustainability (EduS) in primary education be developed?

This phase was mainly an exploratory approach to the issue, which relied on preliminary research on DER and education for sustainability (aimed at children between 6 and 10/12 years old). To this end, there was a literature review and identification/mapping of existing national DER in EduS.

It was also the moment of constitution and organization of the multidisciplinary team that collaborated throughout the whole process, which allowed a network of "critical friends" with diverse expertise to be created. Researchers from different universities and research centers collaborated on this team, bringing together researchers in education,

researchers in science didactics, researchers in digital media, web designers, video game designers, programmers, sound designers, and illustrators. A transdisciplinary approach allowed the development team to achieve a holistic perspective of the problem to be solved, considering multiple factors and perspectives in the design and implementation of the game. Furthermore, a multidisciplinary approach can lead to innovative and creative solutions that would not be possible with a disciplinary approach. Combining different perspectives and knowledge generates new ideas and approaches for game design and development (Lang et al. 2012; Polk 2015). In addition, according to the aforementioned authors, this is an appropriate approach in the context of serious games. Developing serious games requires expertise in various fields, including game design, education, and technology. Therefore, a multidisciplinary approach allows for the creation of more effective, engaging, and entertaining games. This approach also allows for a holistic perspective, which is essential when games address complex issues and challenges (such as EduS).

### 2.3. Phase II—Exploring Possible Solutions

The work carried out in Phase II was based on the identified theoretical frameworks and the knowledge and experience of the various elements of the constituted team. Therefore, this was a privileged moment for discussion and brainstorming in an extended multidisciplinary team, making it possible to explore different ideas for defining the themes, objectives, dynamics, type, and sequence of activities to be included in the DER (in particular, in the "Sustainability at Play" game, the first DER to be developed). This process was dynamic and highly participatory, with an understanding between the various team members and an understanding of the complementarity of their knowledge and experience for the common objective being essential. This participatory design approach among the development team fostered collaboration and co-creation among team members.

In this phase, it was decided that one of the EDRs in EduS would be a serious game that allowed children to explore the ecological footprints associated with the various tasks of their day-to-day—the game "Sustainability at Play," the conception and development process of which is presented in this article. This decision implied, on the one hand, the pursuit of solutions and design proposals for the development of the intended resource and, on the other hand, the constitution of a database with the EF values necessary for the various daily activities that would be considered in the game.

In this conceptualization phase, the team focused on developing the game concept, mechanics, and narrative. During this stage, the team created a game design document that outlined the game's objectives, rules, gameplay mechanics, and requirements. For the team to envision the path that the game would take, creating a flowchart (see Figure 1) was an essential step, as it helped to visualize the game's mechanics, objectives, and progression. In addition, this visualization helped the team identify potential game mechanics issues and ensure that the game was balanced and challenging. A study by Nacke et al. (2009) found that flowcharts effectively surface potential usability issues with games during the design process, which can be addressed through iterative testing and refinement. The flowchart also helped communicate the ideas more effectively to other stakeholders, such as programmers and artists. A study by Lankoski and Björk (2007) found that flowcharts were an effective way of communicating game design concepts to all stakeholders. Using a flowchart ensured that all stakeholders clearly understood the game's objectives and gameplay.

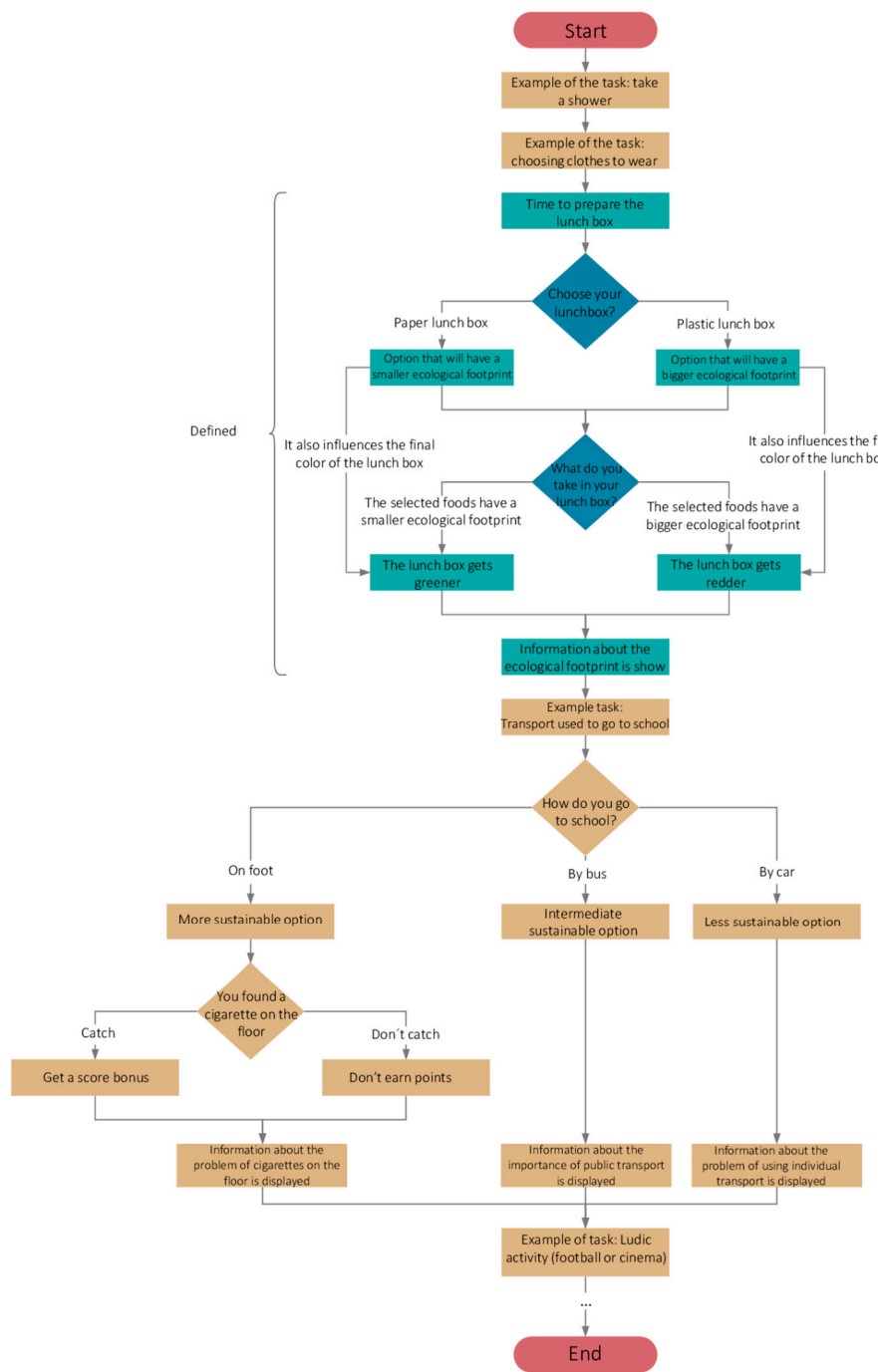

**Figure 1.** Initial gameplay flowchart.

### 2.4. Phase III—Designing and Constructing the DER "Sustainability at Play" Game through Iterative Cycles of Testing and Refinement

The process of the design and construction of the DER "Sustainability at Play" game was, on the one hand, systematic and intentional, following the iterative cycles defined. However, it was also a moment of great creativity and openness for the various team members. In this phase, the ideas generated in the previous moments, generally still vague and not very concrete, gradually become more evident, detailed, and operationalizable. This is the time to consider each of the ideas generated previously and discuss their potential for realization. The team then reduces the initial options to a limited number of possible solutions and collaboratively builds the design skeleton. This phase allows the team to

progressively refine the design while ensuring that the game effectively addresses the educational objectives and the needs of the target audience.

This is often also a prototyping phase. Generally, prototypes are used to try to operationalize the possible solutions, and several prototypes can be tested in the same project. In this project, two prototypes for the DER "Sustainability at Play" game were built and validated. During the design and construction phase of this DER, ideas for the activities were also discussed, with the objectives of the game and the learning it allows being defined. The products to be included in each moment of daily life were identified, and the main navigability characteristics were defined.

The two iterative cycles considered in this study, the main tasks performed in each cycle, and the participants are described below.

### 2.4.1. Cycle 1—Construction and Validation of Prototype 1

During Cycle 1, we proceeded to the construction of Prototype 1. This construction involved defining the navigability characteristics of the DER "Sustainability at Play" game (for example, it was intended that this resource be easy and predictable to navigate, that there should be no navigation failures, and that help mechanisms should be available and strategically located to facilitate access to information).

It also consisted of the elaboration of the initial proposals for the illustrations of each of the scenarios to be considered in the game. At this stage, it was determined that two of the main contexts in which children move in their daily life would be considered: the home and the school. In each of these contexts, it is possible to go through different scenarios, such as the bedroom, the bathroom, the kitchen, and the classroom. Furthermore, in the various scenarios also considered, different products necessary for the children's daily activities were available (for example, clothes, toothpaste, books, and food products). In this cycle, all of these elements were identified, listed, and illustrated, and prototypes were created for the various scenarios included in the game (and all the elements they would have to include).

In this phase, the first version of the database with the EF values of the products to be used in the various scenarios of the DER "Sustainability at Play" game was concluded. This database was built based on several studies on the life cycles and EF of various products, which was a complex and lengthy process of research and literature review and information systematization. The main difficulty in constructing a database useful for the DER "Sustainability at Play" game was finding studies with EF values that would allow for product comparison. The EF is calculated for each product according to a set of variables (e.g., origin, transport, processing, packaging), and it was challenging to obtain comparable EF values (e.g., 1 L of milk in a tetra pack and 1 L of milk in a glass bottle) for the same product (considering the same place of consumption and the same quantity of product).

Still, during this cycle, Prototype 1 was validated. This validation focused on two elements: (i) the database with the EF values and (ii) the initial illustrations made for the several scenarios of the game. To that end, it was necessary to define strategies and methods for the validation and to identify and contact the possible validators.

An expert carried out the validation of the content of the database on PE. This validation occurred in three moments: (i) a first moment, in which the two researchers of the team responsible for the database and the invited expert participated, and the DER "Sustainability at Play" game that was being developed and the structure and purpose of the EF database (the target of validation) were presented; (ii) a second moment of analysis work by the expert; and (iii) a third moment of a new meeting between the researchers and the expert, in which feedback was provided and some suggestions for changes that the expert considered necessary were made.

The illustrations were also validated. The illustrators created the illustrations based on the ideas and information that emerged during the meetings and brainstorming sessions held by the larger multidisciplinary team. The extended multidisciplinary team validated the initial proposals for the illustrations in a joint face-to-face session. This validation

focused on aspects such as the appropriateness of the images to the target audience and the proposed activities, the presence of stereotypes, gender equality, the colors used, and the aesthetics of the game scenarios.

At the end of the validations performed, it was possible to identify the changes to be made to Prototype 1 and to start Cycle 2 and the construction and validation of Prototype 2.

2.4.2. Cycle 2—Construction and Validation of Prototype 2

Cycle 2 consisted of the construction and validation of Prototype 2. This construction started based on the previous cycle, considering the aspects pointed out and discussed in the validation processes and subsequent meetings of the extended multidisciplinary team. Cycle 2 made possible:

- The revision and reformulation of the game structure and the initial prototypes according to the feedback from the various intervening parties (extended multidisciplinary team, specialists, and primary students).;
- The reformulation of part of the illustrations of the basic scenarios of the game;
- The construction of the first complete version of the DER "Sustainability at Play" game, with the new illustrations and the complete scenarios (Figure 2).

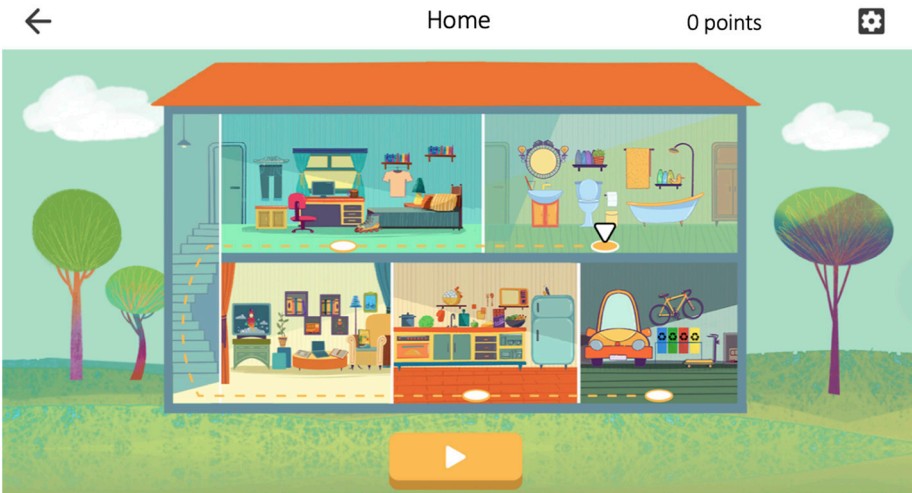

**Figure 2.** DER "Sustainability at Play" game house scenarios.

In the evaluation phase of Cycle 2, the team chose to conduct a pilot study with Prototype 2 and with the collaboration of the game's final users—primary school students. The planning of the pilot study involved defining the methods, strategies, and instruments for data collection and analysis and identifying the participants for the study.

The pilot study of the DER "Sustainability at Play" game was carried out in two sessions, on 5 and 6 April 2022, with two classes in the third year of primary school from the same grouping of schools (designated as Class A and Class B). A total of 20 children was involved—8 children from class A and 12 children from class B. Each session took place in a formal classroom context, occupying a teaching time of 60 min.

These piloting sessions aimed to collect data on the following (attached data collection instrument—Appendix A):

1. The children's understanding of the dynamics of the game itself;
2. The children's perceptions of the layout, navigability, and readability of the game;
3. The knowledge covered during gameplay;
4. The children's emotional state throughout the gameplay.

For items 1 and 2, the data collection instrument presented a three-level agreement scale (totally agree, partially agree, do not agree) for children to position themselves in relation to each of the proposed statements. For item 3, the presented scale had two levels

(agree, do not agree) and for item 4, the children selected, from a previous list, the feelings that best described what they felt while they were playing the game. The scales used were simple given the age of the children who participated in the pilot.

The piloting session was organized into five distinct moments: (1) previous preparation with the teacher responsible for the class in order to explain the dynamics and purposes of the piloting sessions; (2) a brief presentation of the purposes of the piloting session to the children; (3) distribution and explanation to the children of the data collection form about the game; (4) distribution of the tablets by each group, composed of two to three children; and (5) free exploration of the game and, simultaneously, filling in the form by the children.

These sessions took place under the supervision of the teacher responsible for the class to clarify the children's possible doubts. During these sessions, a member of the DER_Sciences team was present online. It should be noted that, for these two sessions with children, informed consent of the parents and of the children themselves was obtained.

After the children had explored the game, the data were analyzed and the following results were obtained.

With regard to the analysis of the data collected during the pilot carried out with the children, almost all the children who participated in the pilot (90%) showed positive values, and more than half of them (55%) showed values higher than 70 points out of 100 in the learning announcements (Table 2).

**Table 2.** Game points achieved by the children.

| Score | Percentage of Children | Number of Children |
|---|---|---|
| 70 to 100 points | 55% | 11 |
| 50 to 69 points | 35% | 7 |
| 0 to 49 points | 10% | 2 |

In all learning statements, more than half of the children responded appropriately (Table 3). The learning statements "Recognizes that the value of the ecological footprint relates to various categories of consumption (e.g., hygiene, clothing, and food)," "Recognizes that the ecological footprint of each product is related to various aspects, such as the material it is made of, its packaging, its provenance . . . ," and "The choices I make in my daily life contribute to increasing or decreasing my ecological footprint" were the learning statements for which children demonstrated the highest scores, reaching values equal to or higher than 90%. The learnings "Everyday choices I make contribute to increasing or decreasing my ecological footprint" and "All products, even 'green' products, have an ecological footprint" were those with which the children demonstrated the most difficulty.

In general, as shown in Table 4, the children showed satisfaction with the game (around 75% of the children indicated being satisfied and very satisfied with the game). In addition, they showed positive feelings/emotions towards this resource, namely, 45% of the children felt excited when playing, 20% were curious, 15% were surprised, and another 15% were bored.

In general, children liked the front end of the game in terms of colors and illustrations, as well as the musical elements (Table 5). We can also verify that the game was adequate for the age group it was aimed at, insofar as the children revealed having understood the proposed tasks. The navigability of the game was also positively evaluated, in terms of both the ease of reading the information on the screen, entering, and exiting the quizzes and returning to the main game.

**Table 3.** Knowledge promoted by the game.

| Apprenticeships in Assessment | | Percentage of Children | | Number of Children | |
|---|---|---|---|---|---|
| | | Agree | Disagree | Agree | Disagree |
| It recognizes that . . . | My ecological footprint is a measure of my lifestyle on the planet. | 85% | 15% | 17 | 3 |
| | The ecological footprint value focuses on several consumption categories (e.g., hygiene, clothing, and food). | 95% | 5% | 19 | 1 |
| | All products, even "green" products, have an ecological footprint. | 55% | 45% | 11 | 9 |
| | Only "yellow" and "red" products have an ecological footprint. | 30% | 70% | 6 | 14 |
| | To reduce my ecological footprint, I should always buy smaller products. | 90% | 10% | 18 | 2 |
| | If I choose the products I like, I contribute to the reduction of my ecological footprint. | 45% | 55% | 9 | 11 |
| | The ecological footprint of each product is related to several aspects, such as the material it is made of, its packaging, its origin . . . | 95% | 5% | 19 | 1 |
| | The same object/product made of different materials (e.g., toothbrushes) can have different effects on the environment. | 75% | 25% | 15 | 5 |
| | The choices I make in my daily life contribute to increasing or decreasing my ecological footprint. | 90% | 10% | 18 | 2 |
| | The ecological footprint is only related to the products we buy in shops. | 40% | 60% | 8 | 12 |

**Table 4.** Children's emotions during the game.

| Feelings about Playing the Game | Percentage of Children | Number of Children |
|---|---|---|
| Surprised | 15% | 3 |
| Curious | 20% | 4 |
| Enthused | 45% | 9 |
| Bored | 15% | 3 |
| Inattentive | 0% | 0 |
| Dissatisfied | 0% | 0 |
| Other | 5% | 1 |

Most of the children said that they understood the relationship between the products they chose, the points they earned, and the size of the green bar. Most children (80%) indicated that they fully understood why products have a frame on them; however, a high percentage (70%) of those did not understand that the green frame of a product also implied that they have an ecological footprint and/or that it was not only products with yellow and red frames that have an ecological footprint. Those who understood the purpose of the bonus quizzes were the ones who used them most often.

This validation found that the game fulfils its entertainment functions, is adequate and easy to navigate, and allows EP content to be explored with children.

**Table 5.** Children's considerations about the game.

| Aspects under Consideration | Percentage of Children | | | Number of Children | | |
|---|---|---|---|---|---|---|
| | Agree | Partly Agree | Disagree | Agree | Partly Agree | Disagree |
| I liked the colors and settings (e.g., bedroom, kitchen, school) of the game. | 85% | 15% | 0% | 17 | 3 | 0 |
| I could easily read what was on the screen. | 55% | 40% | 5% | 11 | 8 | 1 |
| The information that appeared on the screen (e.g., the name of the products, the summary, the icons) helped me to advance in the game. | 80% | 20% | 0% | 16 | 4 | 0 |
| I liked the music that came with the game. | 60% | 25% | 15% | 12 | 5 | 3 |
| The instructions given to me were helpful for playing. | 85% | 15% | 0% | 17 | 3 | 0 |
| It was easy to enter and exit the bonus quizzes and go back to where I was. | 65% | 30% | 5% | 13 | 6 | 1 |
| I realized that the green bar that accompanies every scenario represents the credit points I have for playing. | 90% | 10% | 0% | 18 | 2 | 0 |
| I understood why a color (green, orange or red) appears to frame each product I choose. | 80% | 15% | 5% | 16 | 3 | 1 |
| I understood the relationship between the products I choose, the points I earn, and the size of the green bar. | 70% | 30% | 0% | 14 | 6 | 0 |
| I read and understood the summary at the end of each scenario. | 70% | 20% | 10% | 14 | 4 | 2 |
| I understood what the bonus quizzes were for. | 65% | 20% | 15% | 13 | 4 | 3 |
| I took advantage of all the bonus quizzes to earn extra points. | 55% | 35% | 10% | 11 | 7 | 2 |
| I understood that when the green bar runs out the game ends. | 85% | 10% | 5% | 17 | 2 | 1 |
| After completing all the scenarios or after I had used up the green bar, I replayed the game to try and improve my score. | 60% | 30% | 10% | 12 | 6 | 2 |

### *2.5. Phase IV—Reflection*

The last phase of the EDR approach is a phase of reflection on the process and the resulting product. During this phase, the team reflects on the entire process, analyzing what worked well and what could have been improved. This reflection allows for continuous improvement and learning.

As regards the path, the research methodology followed proved to be appropriate for the objective set and allowed the original research question to be answered. The use of the EDR approach enabled the design and development of a DER in EduS for the early years of schooling, within a multidisciplinary team and through a collaborative and iterative process open to the participation of different partners (researchers, experts, and primary school students). This was a process based on reference literature, but it was also built on the experience and competence of the team members.

The construction and use of prototypes allowed for the realization of the initial ideas and their improvement, as well as the collaboration of different partners throughout the process. The two cycles implemented allowed for the collection of data and the reformulation of the DER according to the feedback obtained, making the final result more founded and adequate for both the initial objectives of the team and the interests and needs of its end users.

The final result, the DER "Sustainability at Play" game, is briefly described in the Results section, resulting from the process and the various reformulations that were carried out.

## 3. Results

The result of the implemented EDR approach is the DER "Sustainability at Play" game. In this game, children are asked to go through a narrative of a day in six different scenarios at home (bathroom, bedroom, kitchen, garage, living room) and at school (classroom). There is a total of nine everyday situations in the different scenarios, with a total of 20 situations to choose from. The player is challenged, in the different scenarios, to choose products and goods (toothpaste, jeans, milk, books), which they consume at different times of their day (dressing, hygiene, meals, traveling). As initial support, hints are provided to help players understand the dynamics and the objective of the game in question, as illustrated in Figure 3. These hints are provided in the form of textual and visual cues and are carefully designed to guide players through the gameplay and ensure that they are able to engage with the educational content in an effective manner. By providing these hints, the team aimed to create a positive user experience and increase the likelihood of players successfully completing the game and retaining its takeaway messages.

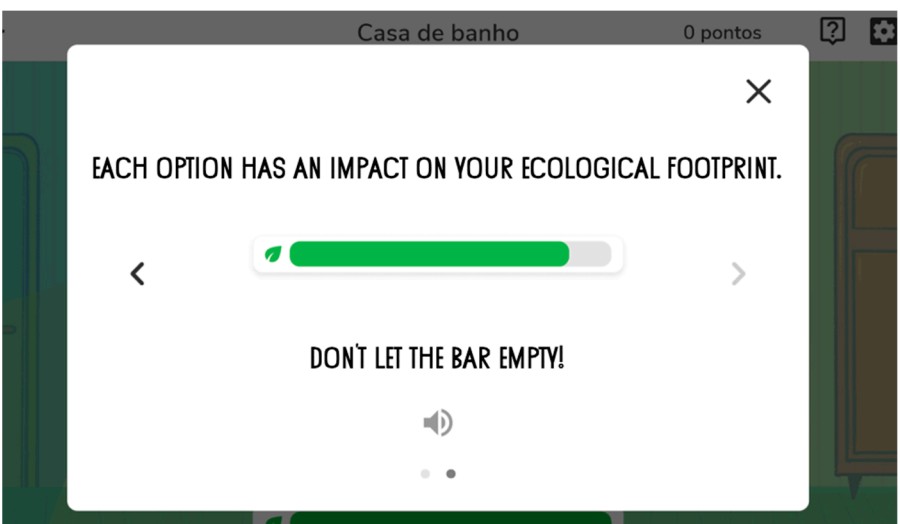

**Figure 3.** Game situation: Choose a toothpaste.

These naturally had different impacts on the bar associated with the EF the player has available (Figure 4). Depending on the player's choices and the smaller or larger the ecological footprint, the player accumulates points and the EF bar decreases. The game is interrupted and ends if the player exhausts the EF bar.

Three alternatives are presented for each good and product proposed, each with a different associated PE value represented through a color system (green—low EF, orange—medium EF, and red—high EF). After the player's choice, the alternative appears with the outline corresponding to the PE value (Figure 5). Thus, depending on the choice of product and goods the player makes in each game scenario, the bar associated with the PE decreases more slowly or more quickly.

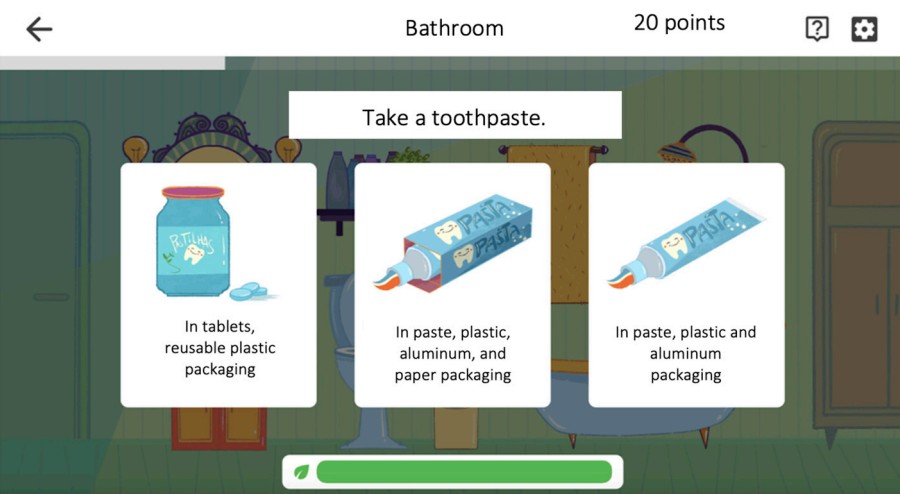

**Figure 4.** Game situation: Choose a toothpaste.

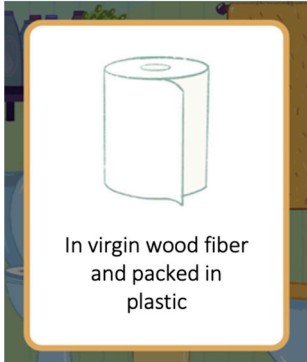

**Figure 5.** Game situation: Choose an object with a medium ecological footprint.

In the end, the player is presented with a summary of their choices in each scenario (Figure 6), where, through the information provided to them about each of the products and goods they choose, the player can identify how everyday choices contribute to managing the EF in order to have as little impact as possible (e.g., the material and shape of the packaging, the place of origin, whether the food is in season or not).

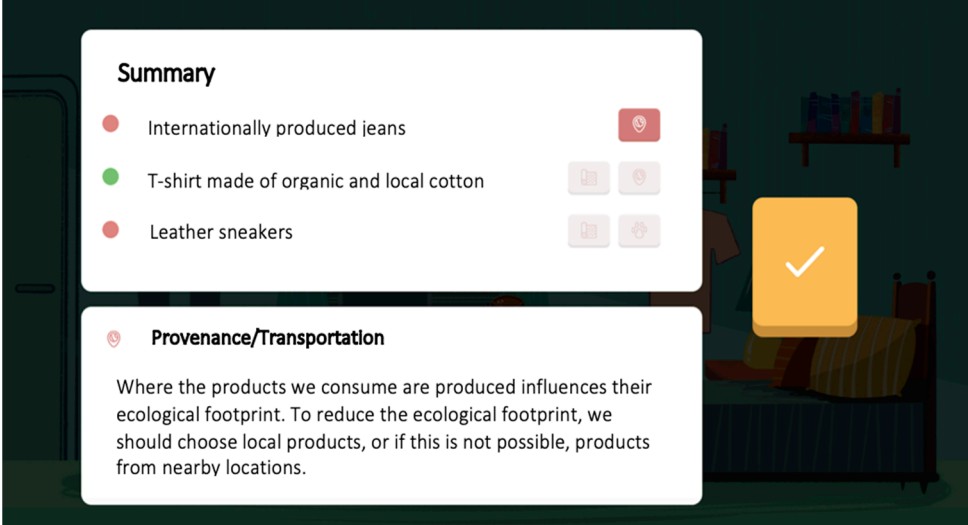

**Figure 6.** Game situation: final summary of a scenario.

In order to increase their score, at the end of each scenario the player is challenged to answer the bonus quiz, which they can accept or decline. The bonus quiz explores extra information (e.g., water footprint, household solid waste separation, the life cycle of a particular product) about certain objects/goods available in that scenario (Figure 7).

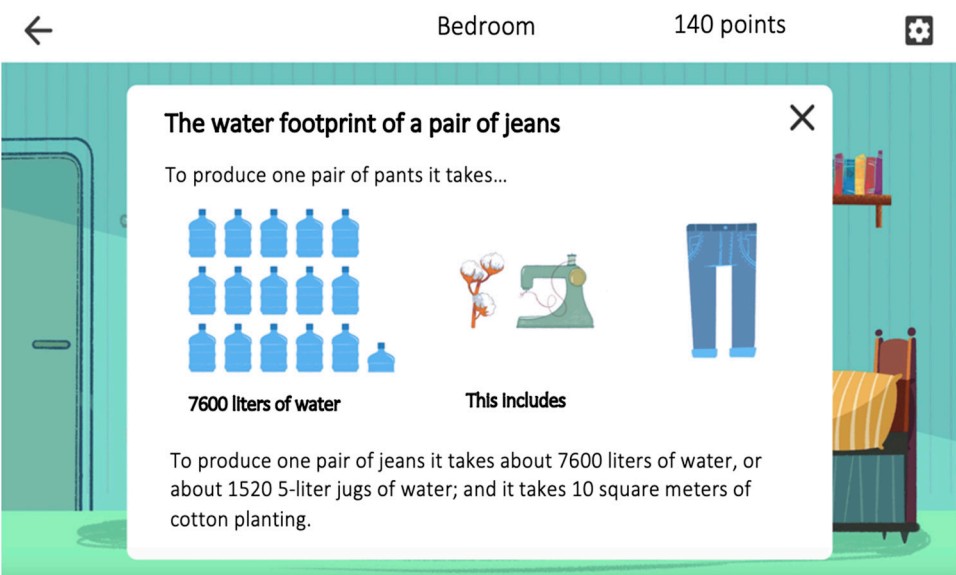

**Figure 7.** Game situation: bonus quiz information.

At the end of the game, a summary of the score of the player's choices, the bonus quizzes, and the final score appears (Figure 8).

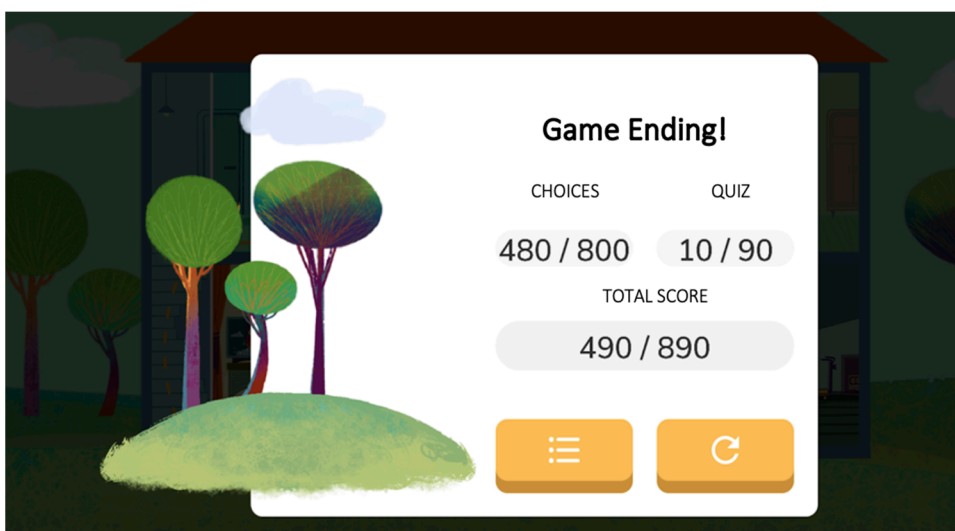

**Figure 8.** Game situation: end game screen.

## 4. Discussion

The discussion about the work developed and presented here is essentially based on two structuring aspects of this article: the development process of the presented DER, using a DBR methodology, and the game "Sustainability at Play," which is the final result.

The option for a DBR methodology for the design and development of a digital educational resource to address EduS issues in the early years of schooling was based on the reference literature. After a careful literature review, the team realized that this would be the most appropriate development methodology. In fact, the literature consulted suggested that the education design research methodology can be effective in the development of

digital educational resources, providing a viable alternative to traditional educational research approaches (Lehtonen 2021; Liu and Yang 2010; McKenney and Reeves 2019; Reeves 2006; Rozo and Real 2019; Turucz et al. 2021; Xie et al. 2018).

Some of the main characteristics of this methodology were decisive for the success of the process developed and the resource obtained, highlighting:

(i) Iterativeness. Since iterativeness is one of the main characteristics of EDR, the iterative cycles considered in the development of the "Sustainability at Play" game were fundamental to gradually designing and developing an innovative educational resource, based on research results and suitable for its real context of use (classroom). The sequence of these cycles made it possible to conceive, test, and improve the resource that was intended to be developed, showing the adequacy of the chosen methodology to the task that the team had at hand. Several authors, through the results obtained in their studies, showed that this methodology is one of the most suitable for the development of educational resources. For example, Lehtonen (2021) proposed a design framework and guidelines for conducting educational design research that can assist in developing educational technologies that ensure their educational benefits, feasibility, and successful real-world utilization and adoption.

(ii) Multiple participants and different validations. The involvement of several participants in the different phases of the EDR also proved to be an added value for the development of the resource and for its adequacy for the context of use. Different participants (researchers, experts, teachers, students) may be involved in different stages of the process, based on the needs and difficulties that arise (McKenney and Reeves 2019; Herrington et al. 2007). Reeves (2006) even talked about the social responsibility present in the whole process. We emphasized, in this regard, the participation of experts and primary school students. The experts, based on their knowledge, validated scientific content and ways of operationalizing it, and students acted as the end users and provided feedback regarding the attractiveness of the interface, navigability, interest, and appropriateness of the activities. Campo and Rodríguez-Abitia (2018), in their study, found that digital educational resources with high levels of intuitiveness and attractiveness can lead to more effective learning.

(iii) Multidisciplinary collaboration. The multidisciplinary collaboration was, from the outset, an asset to the entire process. Regular meetings with various team members (both face-to-face and online) allowed close communication to be maintained between the researchers and boosted the development of the work, always from a perspective of broad multidisciplinary interaction. It was this collaborative and deeply participative dynamic that allowed the challenges that emerged throughout the process to be overcome, both in the design of the resource and in its operationalization.

Overall, and regarding the development process, our results seem to confirm what the literature suggests: that the EDR methodology can lead to the development of high-quality DERs that are innovative and appropriate for different contexts based on the relevance and effectiveness of the methodology (followed for the development of the DER "Sustainability at Play" game). At the methodological level, the use of a pre- and post-test, as in other studies (Veronica and Calvano 2020; Oliveira et al. 2021), could assess the contribution of the game in a more effective way. This is, methodologically speaking, one of the limitations of this study.

As regards the DER "Sustainability at Play" game, and as found in other studies (e.g., Chappin et al. 2017; Jesus et al. 2021; Leal et al. 2022; Oliveira et al. 2021; Vestena and Bem 2020), this particular game seems to meet its didactic and entertainment purposes. The validation of the game with the target audience through the data collection instrument allowed us to verify that its design is adequate for the defined objectives and that its gameplay is appreciated by the users (e.g., the children revealed to understand the game's objective and had positive feelings towards its gameplay). The use of similar data collection tools allowed, in other studies, criteria to be verified under evaluations such as graphics, content, language, and interaction, as well (Leal et al. 2022; Oliveira et al. 2021). The DER

"Sustainability at Play" game can be considered a digital educational resource because it meets pedagogical–didactic principles, namely, those stated by Cardoso et al. (2022):

1. Creating meaningful contexts for learning. The game places the player in a scenario similar to what their daily life could be like, involving them in a real and immersive learning context.

2. Curricular integration. The game fits into the 4th grade Essential Learning in Environmental Studies (Portuguese curriculum guidelines for the teaching of science in the early years of schooling—http://www.dge.mec.pt/estudo-do-meio, accessed on 29 June 2023), namely, in the learning statement "Relate the increase in world population and consumption of goods with changes in the quality of the environment (destruction of forests, pollution, resource depletion, extinction of species, etc.), recognizing the need to adopt individual and collective measures to minimize the negative impact" (p. 10). This aspect was also verified in the development of other games of this nature (Oliveira et al. 2021).

3. Implementation of a constructivist-based design. The game design contemplates challenges that involve the student in the learning process.

4. Promotion of the student's autonomy. The instructions that appear throughout the game ensure that the child can play autonomously. This does not mean that the game cannot be explored with strategic guidance from the teacher or another adult.

5. Promoting the involvement and motivation of the pupil in the teaching and learning process. The playful character of the game and the nature of the activities proposed in it foster the active involvement and motivation of the students.

According to Oliveira et al. (2021), the game's ability to address complex topics such as sustainability while remaining relatable to children's reality and context is highly appreciated. By placing the child as the protagonist, the game becomes meaningful and possesses the potential to raise awareness and encourage the adoption of effective attitudes towards consumption.

The score system, EP bar, and audio were identified as limitations of the game despite the possibility of accumulating points with additional quizzes, as in other game examples (Veronica and Calvano 2020; Leal et al. 2022). Providing significance to the accumulated score was paramount, and one approach to achieving this was by offering players the opportunity to purchase and plant trees. This action allowed players to witness the direct impact of their EP as the number of trees increased. Players were encouraged to reflect on their EP and its positive contribution to the environment by connecting the score to tangible actions like tree planting, such as in the game "Produtos Perigosos ou Sustentáveis" (Oliveira et al. 2021), which allows the use of the points earned in hints or the possibility of several answer attempts. In this game, the EP bar determines the player's life, and there is no possibility of increasing it. On the other hand, all the statements should be accompanied by an audio to ensure that potential difficulty with reading for the player does not prevent them from playing, an aspect that Souza et al. (2020) also pointed out as a limitation to the game "Universal Machine Ecological: U.M.E." (Souza et al. 2020).

Despite these limitations, in a global way, the results point to the fact that this is a digital game with enough educational potential to promote education for sustainability in the early years of schooling.

## 5. Conclusions

This paper intended to present and describe the methodology used to develop the DER "Sustainability at Play" game, an educational resource aimed at primary education.

For the development of this educational resource, a multidisciplinary team was organized and the EDR approach was followed. The implementation of this approach was organized in four phases: Phase I—Problem Analysis, Phase II—Exploration of possible Solutions, Phase III—Design, and Phase IV—Reflection. During Phase III, two iterative cycles were implemented, enabling the prototyping of the DER and the collaboration of several participants (e.g., researchers, illustrators, web designers, experts, primary school

students) in the development, validation, piloting, and restructuring of this educational resource. Some of the main features of this methodology—iterativeness, multiple participants, different validations, and multidisciplinary collaboration—were absolutely essential to the final result, evidencing the relevance, suitability, and effectiveness of the EDR approach for the development of these kinds of educational resources. Thus, the procedures followed and the results obtained seem to be aligned with the reference literature: EDR can lead to the development of high-quality DERs that are innovative and appropriate to different contexts, which proves it to be a privileged methodology for the development of DERs such as the "Sustainability at Play" game.

Additionally, the DER "Sustainability at Play" game allows us to explore with children ways of intervening to reduce the EF inherent in daily choices and, consequently, their impact. It also helps raise awareness of the fact that all choices have an impact—some more than others—and that it is important to balance our consumption without giving anything up.

It would be interesting, in a future investigation, to have the opportunity to evaluate the implementation of the DER "Sustainability at Play" game with broader and more diversified groups of participants involving, for example, students from other levels of education and teachers from different areas, and extend this assessment to non-formal teaching and learning contexts (e.g., the family context). The use of this methodology for the development of new DERs, following the steps and procedures presented, would also be a privileged research opportunity, making it possible to add knowledge about the adequacy of the EDR for the development of this type of resource. Nevertheless, the development of a DER for education for sustainability in primary education, made available by the Ministry of Education on an open platform and that resulted from the collaboration between different research centers and multiple participants, is an added value of the presented resource, allowing for the union of collaboration, research, and innovation.

In summary, due to the privileged characteristics of the development process (cyclical and iterative process, involvement of different partner institutions, multidisciplinary team, available development period, expert validations, piloting, and availability of the resource on an open-access platform); the proposed activities, which are considered innovative both from the point of view of the themes to be addressed and from the didactic point of view; and the intended audience (primary school children), this is undoubtedly an important contribution to the promotion of the SDGs in Portugal, especially SDG4.

**Author Contributions:** Conceptualization, P.S.; methodology, P.S.; validation, P.S. and J.P.; formal analysis, P.C.S. and J.P.; investigation, P.S., J.P., A.F. and A.V.R.; resources, P.S., J.P., A.F. and A.V.R.; data curation, P.S., J.P. and A.V.R.; writing—original draft preparation, P.S. and P.C.S.; writing—review and editing, P.S., P.C.S., J.P., A.F. and A.V.R.; project administration, A.V.R.; funding acquisition, A.V.R. All authors have read and agreed to the published version of the manuscript.

**Funding:** This work was financially supported by (i) POCH-04-5267-FSE-000124; (ii) national funds through FCT—Fundação para a Ciência e a Tecnologia I.P. and University of Aveiro in the scope of the framework contract foreseen in numbers 4, 5, and 6 of article 23 of Decree-Law 57/2016, of 29 August, changed by Law 57/2017 of 19 July, and (iii) national funds through FCT—Fundação para a Ciência e a Tecnologia through the Programa Operacional Centro I.P. under grant SFRH/BD/143370/2019.

**Institutional Review Board Statement:** The study was conducted in accordance with the procedures required to guarantee the confidentiality and anonymization of the children who participated in this study. It is not possible, at any time, to identify any of the children. The required authorizations for the children's participation during the validation phase were granted by their parents, who gave their written informed consent.

**Informed Consent Statement:** Informed consent was obtained from all subjects involved in the study.

**Conflicts of Interest:** The authors declare no conflict of interest.

## Appendix A

### ˮMY day: SUSTaInaBILITY aT PLaYˮˮ

Hello! On Periscope Island, the little monsters that live there have created a game, but this one cannot be complete without your collaboration!
Help them complete it by giving your honest opinion about the game! So let's go!

| I mark with an x my degree of agreement with the following statements: | Totally agree ☺ | Partially agree 😐 | Do not agee ☹ |
|---|---|---|---|
| I liked the colours and settings (e.g. bedroom, kitchen, school) of the game | | | |
| I could easily read what was on the screen | | | |
| The information that appeared on the screen (e.g. the name of the products, the summary, the icons) helped me to advance in the game | | | |
| I liked the music that came with the game | | | |
| The instructions given to me were helpful to play | | | |
| It was easy to get in and out of the bonus quizzes and back to where I was | | | |
| I realised that the green bar that accompanies every scenario represents the credit points I have for playing | | | |
| I understood why a colour (green, orange or red) appears to frame each product I chose | | | |
| I understood the relationship between the products I choose, the points I earn and the size of the green bar | | | |
| I read and understood the summary at the end of each scenario | | | |
| I understood what the bonus quizzes were for | | | |
| I took advantage of all the bonus quizzes to earn extra points | | | |
| I understood that when the green bar runs out the game ends | | | |
| After completing all the scenarios or after I had used up the green bar, I replayed the game to try and improve my score | | | |

| I mark with an x my degree of agreement with the following statements, taking into account what I learned from the game. | Agree ☺ | Do not agree ☹ |
|---|---|---|
| My ecological footprint is a measure of my lifestyle on the planet | | |
| The ecological footprint value focuses on several consumption categories (e.g. hygiene, clothing and food) | | |
| All products, even "green" products, have an ecological footprint | | |
| Only 'yellow' and 'red' products have an ecological footprint | | |
| To reduce my ecological footprint, I should always buy smaller products | | |
| If I choose the products I like, I contribute to the reduction of my ecological footprint | | |
| The ecological footprint of each product is related to several aspects, such as: the material it is made of, its packaging, its origin... | | |
| The same object/product made of different materials (e.g. toothbrushes) can have different effects on the environment | | |
| The choices I make in my daily life contribute to increase or decrease my ecological footprint | | |
| The5 ecological footprint is only related to the products we buy in the shops | | |

I mark with an x how I felt while playing.

☺ Surprised     ☺ Bored
☺ Curious       😐 Inattentive
☺ Enthused      ☹ Dissatisfied
☺ Other _______________

I paint the stars according to my degree of overall satisfaction with the game.

Not satisfied ☆
Little satisfied ☆ ☆
Satisfied ☆ ☆ ☆
Very satisfied ☆ ☆ ☆ ☆
Completely satisfied ☆ ☆ ☆ ☆ ☆

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
