# Peer review of "Sustainability at Play: Educational Design Research for the Development of a Digital Educational Resource for Primary Education"

_socsci, doi:10.3390/socsci12070407_

Round 1

Reviewer 1 Report

This manuscript overall has a good quality, author can improve several section:
1. Introduction can be improve by insert several sentences about urgency of this research

2. Discussion section can be improve by authors by comparing this media with another media

can be improve with proofreaders services

Author Response

We would like to thank the reviewers for carefully reading our manuscript and for giving such constructive comments in order to help improving the quality of the paper. In the revised version of the manuscript we have tried to consider all the points that were raised. These are repeated below. Text passages that have been added to the manuscript are given in italics.

Point 1. This manuscript overall has a good quality, author can improve several section:

 - Introduction can be improve by insert several sentences about urgency of this research

Response 1: Following the reviewer's suggestion, we added some additional information in the Introduction regarding the urgency and relevance of this research:

According to Napal, Mendióroz-Lacambra and Peñalva (2020), one of the fields that more urgently needs to utilize the potential benefits of digital technologiesICT to transform learning is sustainability, evidencing the pertinence of studies such as the one presented here. In particular, several authors emphasize the relevance of serious games in an EduS approach (Neset, Andersson, Uhrqvist and Navarra 2020; Ouariachi, Olvera-Lobo, Gutiérrez-Pérez, 2019; Stanitsas, Kirytopoulos and Vareilles, 2019). Serious games are important since they allow players to experience unfamiliar circumstances that are not possible in real life, enabling to raise awareness and promote changes in attitude and behavior in players (Ouariachi, Olvera-Lobo, Gutiérrez-Pérez, 2019; Stanitsas, Kirytopoulos and Vareilles, 2019). Neset, Andersson, Uhrqvist and Navarra (2020) emphasize the importance of these resources in the teaching and learning process, stating that serious games can support teachers to strengthen their EduS as it provides an experience of climate adaptation, based on systems thinking and action orientation. 

The growing acknowledgment of EduS and the perceived benefit of serious games for teaching students about sustainability are both recognized arguments that highlight the relevance of this field of study (Hallinger, Wang, Chatpinyakoop, Nguyen, and Nguyen, 2020).

Point 2. Discussion section can be improve by authors by comparing this media with another media.

Response 2: We agree with the reviewer’s remarks. In order to consider this suggestion, we add some more information in the Discussion:

As regards the DER "Sustainability at Play" game, and as found in other studies (e.g. Chappin, Bijvoet and Oei, 2017; Jesus, Silveira, Araújo, and Penha 2010; Leal, Ramos and Ramos 2010; Oliveira, Filho, Guilherme and Leme 2021; Vestena and Bem 2010), this particular game seems to meet its didactic and entertainment purposes. The validation of the game with the target audience, through the data collection instrument questionnaire, allowed us to verify that its design is adequate to the defined objectives and that its gameplay is appreciated by the users (e.g. children revealed understanding the game's objective and having positive feelings towards its gameplay). The use of similar this data collection tools allowed, in other studies, to verify criteria also under evaluation such as graphics, content, language and interaction (Leal, Ramos and Ramos 2022; Oliveira, Filho, Guilherme and Leme 2021).

Reviewer 2 Report

The abstract presented has an unstructured style, although the article corresponds to a possible scientific publication. For this reason, it is suggested to reformulate the summary so that it includes the objective, method, results and the most important conclusions.

Regarding the objective of the research:

Stated objective in summary: ´present the development process of this DER’

Objective declared in introduction: 'to present and describe the methodology used to develop the DER game "Sustainability at Play"´

Objective declared in the method: 'to design and develop EHR to work on issues related to sustainability in the first years of schooling'.

It is suggested to use the same wording in the construction of the objective. Not having a clear and defined objective can generate confusion and invalidate the investigation.

The stated research question is: ‘How to develop Digital Educational Resources (DER) in Education for Sustainability (EduS) in the early years?’.

It is important to remember that the research question is the central question that must be answered by the investigation. It helps to clearly define the direction that the study should follow. This question usually addresses the problem or issue, which, through data analysis and interpretation, is answered at the end of the study.

In this case, the research question is not related or coherent with the objective proposal, which reduces the consistency of the entire work. It is suggested to reformulate the approach made to give initial and constant coherence to the work and to be able to develop a systematized proposal.

Regarding Method, it is important to remember that this section must include sufficient detail so that others can replicate, understand and use the results obtained from the proposed research. The approach of this study makes an effort to describe in detail what was done. However, it includes limited information about the population, sample, temporality, etc., which makes it difficult to understand how the pilots were carried out.

The results should “provide a concise and accurate description of the experimental results, their interpretation, as well as the experimental conclusions that can be drawn” (https://www.mdpi.com/journal/socsci/instructions). Although the objective of the study would be aimed at presenting and describing the methodology used to develop the DER game “Sustainability at Play”; the results must clearly show that the tests applied on the corresponding samples were obtained during the entire time that the development of the game took.

The discussion takes the results obtained and compares them with other studies, theories, etc., that are located in the scientific literature. If you do not have a clear idea of ​​the results, it will be difficult to raise an adequate discussion.

Quotations from different authors (by placing them in parentheses) should take care of the writing style in each presentation.

Of the total number of references, only 9 correspond to the last 5 years and are properly located. It is suggested to considerably improve the percentage of references of scientific articles on the subject, whose age is located in the last 5 years.

Take great care in the presentation of the references used. The presentation style of each of them must include the necessary and sufficient elements for its location.

Author Response

We would like to thank the reviewers for carefully reading our manuscript and for giving such constructive comments in order to help improving the quality of the paper. In the revised version of the manuscript we have tried to consider all the points that were raised. These are repeated below. Text passages that have been added to the manuscript are given in italics.

Point 1. The abstract presented has an unstructured style, although the article corresponds to a possible scientific publication. For this reason, it is suggested to reformulate the summary so that it includes the objective, method, results and the most important conclusions.

Response 1: We have changed it taking the reviewer’s suggestions into account.

Point 2. Regarding the objective of the research:

Stated objective in summary: ´present the development process of this DER’

Objective declared in introduction: 'to present and describe the methodology used to develop the DER game "Sustainability at Play"´

Objective declared in the method: 'to design and develop EHR to work on issues related to sustainability in the first years of schooling'.

It is suggested to use the same wording in the construction of the objective. Not having a clear and defined objective can generate confusion and invalidate the investigation.

Response 2: We would like to thank the reviewer for pointing these out. The wording of the objective was reviewed throughout the document and standardized, always appearing in the same way in this new version. The final version of the objective formulation is: “to present and describe the methodology used to develop the Digital Educational Resource (DER) “Sustainability at Play” game”.

Point 3. The stated research question is: ‘How to develop Digital Educational Resources (DER) in Education for Sustainability (EduS) in the early years?’.

It is important to remember that the research question is the central question that must be answered by the investigation. It helps to clearly define the direction that the study should follow. This question usually addresses the problem or issue, which, through data analysis and interpretation, is answered at the end of the study.

In this case, the research question is not related or coherent with the objective proposal, which reduces the consistency of the entire work. It is suggested to reformulate the approach made to give initial and constant coherence to the work and to be able to develop a systematized proposal.

Response 3: In an attempt to clarify the various formulations for the objectives presented throughout the text and the lack of consistency noted between the research question and the referred objectives, we would like to explain that the objective of the study carried out was to 'to design and develop DER to work on issues related to sustainability in the first years of schooling', but that the objective of the article would be "to present and describe the methodology used to develop the Digital Educational Resource (DER) “Sustainability at Play” game” (one of the DER developed within the scope of the aforementioned study). This distinction was not well achieved in the submitted text, for which we are grateful for the reviewer's remarks.

In order to improve the coherence between the research question and the presented objective, we chose to slightly change the formulation of the research question - so that it is now formulated as follows “How to develop DER for EduS in Primary Education?” -  and maintain only one formulation for the objective (referred to in the previous point). We expect that these changes have contributed to improving the consistency of the work.

Point 4. Regarding Method, it is important to remember that this section must include sufficient detail so that others can replicate, understand and use the results obtained from the proposed research. The approach of this study makes an effort to describe in detail what was done. However, it includes limited information about the population, sample, temporality, etc., which makes it difficult to understand how the pilots were carried out.

Response 4: We appreciate the comments and suggestions made by the reviewer on this section. However, we are not sure that we fully understood what information could be changed and/or added to make the text clearer in relation to the mentioned aspects.

The description of the pilots includes information about when they were carried out, the number of participants, the location and the instruments used for data collection.

The pilot study of the DER “Sustainability at Play” game was carried out in two sessions, on 5 and 6 April 2022, with two classes in the 3rd year of primary school of the same Grouping of Schools (designated as Class A and Class B). A total of 20 children were involved - 8 children from class A and 12 children from class B. Each session took place in a formal classroom context, occupying a teaching time of 60 minutes.

We also added information about the period in which the development of the DER took place (Table 1) and the constitution of the multidisciplinary team.

Point 5. The results should “provide a concise and accurate description of the experimental results, their interpretation, as well as the experimental conclusions that can be drawn” (https://www.mdpi.com/journal/socsci/instructions). Although the objective of the study would be aimed at presenting and describing the methodology used to develop the DER game “Sustainability at Play”; the results must clearly show that the tests applied on the corresponding samples were obtained during the entire time that the development of the game took.

 The discussion takes the results obtained and compares them with other studies, theories, etc., that are located in the scientific literature. If you do not have a clear idea of ​​the results, it will be difficult to raise an adequate discussion.

Response 5: The methodology of the study that is presented is of a qualitative nature, having been our objective to focus on the description of the process and procedures followed, as well as to analyze the data in an interpretative perspective. The validations and pilots carried out throughout the DER development process are shown in Table 1 and are described in more detail throughout the text. We did not quite understand the following comment: "the results must clearly show that the tests applied on the corresponding samples were obtained during the entire time that the development of the game took", not realizing what could be clarified in the text.

Point 6. Quotations from different authors (by placing them in parentheses) should take care of the writing style in each presentation.

Of the total number of references, only 9 correspond to the last 5 years and are properly located. It is suggested to considerably improve the percentage of references of scientific articles on the subject, whose age is located in the last 5 years.

Take great care in the presentation of the references used. The presentation style of each of them must include the necessary and sufficient elements for its location.

Response 6: We would like to thank the reviewer for bringing this to our attention. The list of references has been updated and more recent references have been added. Also, the reference list was reviewed according to MDPI style.

Reviewer 3 Report

The study focuses on the educational design research for the development of a digital educational resources targeting primary education. The topic is certainly interesting and the structure and flow of the text as well the level of English are satisfactory and only minor corrections are required.

In the abstract, the main aims of the article should be presented earlier in the text. Additionally, it would be helpful to also highlight the novelty of the study and its main contributions.

The introduction section clearly presents the required information. Despite this fact, more references should be included in the theoretical part. For example, the statement in lines 38-39 must be supported by a related reference. Moreover, the authors go over several other related studies and clearly present them and their findings. I believe that the authors should briefly mention why the specific studies were selected instead of others. Was it only the language or were there additional factors? The aims and contributions of the study are clearly presented.

Section 2 is well structured and all the required information is presented in a clear and comprehensible manner while also referencing a satisfactory number of studies. Nonetheless, it is worth noting that there is a clear lack of recent studies (last 5-6 years) being referenced. I believe that this part can be further improved. Table 1 is really helpful to get a better understanding of the development process and its four phases. Tables 2-5 also present important for the study information. The authors should also include the frequency values of each element in Tables 2-5 instead of simply mentioning the percentage. Additionally, I wonder why in Table 5 the only options are “Agree”, “Partly agree”, and “Disagree”? I believe that the authors should provide some additional details.

The result section clearly presents the outcomes. Although the figures used are helpful, I believe that it would be better if the text within them could be translated into English to reach a broader readership.

In the discussion section, the authors must add more references and make connections and comparisons between the findings of other related studies and theirs.

In contrast to the rest of the manuscript, the conclusion section seems a bit lacking. I believe the authors should try to provide more details and conclusive remarks about their findings since several aspects have been analyzed. Some additional comments on the overall limitations should be integrated and suggestions for future research directions should be better highlighted.

As a final remark, I believe that the overall quality of the manuscript is good but I believe that it can be further improved by addressing the above-mentioned comments.

The level of English is satisfactory and only some minor mistakes that exist must be corrected.

Author Response

We would like to thank the reviewers for carefully reading our manuscript and for giving such constructive comments in order to help improving the quality of the paper. In the revised version of the manuscript we have tried to consider all the points that were raised. These are repeated below. Text passages that have been added to the manuscript are given in italics. 

Point 1. The study focuses on the educational design research for the development of a digital educational resources targeting primary education. The topic is certainly interesting and the structure and flow of the text as well the level of English are satisfactory and only minor corrections are required.

Response 1: We appreciate and thank the reviewer’s kind words.

Point 2. In the abstract, the main aims of the article should be presented earlier in the text. Additionally, it would be helpful to also highlight the novelty of the study and its main contributions.

Response 2: We would like to thank the reviewer for pointing these out. The suggested corrections have been made.

Point 3. The introduction section clearly presents the required information. Despite this fact, more references should be included in the theoretical part. For example, the statement in lines 38-39 must be supported by a related reference. Moreover, the authors go over several other related studies and clearly present them and their findings. I believe that the authors should briefly mention why the specific studies were selected instead of others. Was it only the language or were there additional factors? The aims and contributions of the study are clearly presented.

Response 3: The suggested corrections have been integrated in the new version of the text. More references were included in the theoretical part and the referred statement was supported by references. The text was also amended in order to clarify the justification for choosing the presented studies. 

Point 4. Section 2 is well structured and all the required information is presented in a clear and comprehensible manner while also referencing a satisfactory number of studies. Nonetheless, it is worth noting that there is a clear lack of recent studies (last 5-6 years) being referenced. I believe that this part can be further improved. Table 1 is really helpful to get a better understanding of the development process and its four phases. Tables 2-5 also present important for the study information. The authors should also include the frequency values of each element in Tables 2-5 instead of simply mentioning the percentage. Additionally, I wonder why in Table 5 the only options are “Agree”, “Partly agree”, and “Disagree”? I believe that the authors should provide some additional details.

Response 4: More recent studies have been added and incorporated into the Discussion section. The referred tables were also changed according to the suggestions given by the reviewer. An explanation of the parameters of the scale used was added to the text:

For items 1 and 2, the data collection instrument presents a 3-level agreement scale (totally agree, partially agree, do not agree) for children to position themselves in relation to each of the proposed statements. For item 3, the presented scale has 2 levels (agree, do not agree) and for item 4, the children selected, from a previous list, the feelings that best described what they felt while they were playing the game. The scales used are simple given the age of the children participating in the pilot.

Point 5. The result section clearly presents the outcomes. Although the figures used are helpful, I believe that it would be better if the text within them could be translated into English to reach a broader readership.

Response 5: We welcome the reviewer's comments on the Results section. The suggestions made were integrated into the text.

Point 6. In the discussion section, the authors must add more references and make connections and comparisons between the findings of other related studies and theirs.

Response 6: We agree with this remark and include more and more updated references in the Discussion section.

Point 7. In contrast to the rest of the manuscript, the conclusion section seems a bit lacking. I believe the authors should try to provide more details and conclusive remarks about their findings since several aspects have been analyzed. Some additional comments on the overall limitations should be integrated and suggestions for future research directions should be better highlighted.

Response 7: We appreciate the comments made about this section. The text was reformulated in an attempt to integrate the suggestions left by the reviewer.

This paper intended to present and describe the methodology used to develop the DER “Sustainability at Play” game, an educational resource aimed at Primary Education. 

For the development of this educational resource, a multidisciplinary team was organized and the EDR approach was followed. The implementation of this approach was organized in 4 Phases: Phase I - Problem Analysis; Phase II - Exploration of possible solutions; Phase III - Design and; Phase IV - Reflection. During Phase III, two iterative cycles were implemented, enabling the prototyping of the DER and the collaboration of several participants (eg. researchers, illustrators, web designers, experts, primary school students) in the development, validation, piloting and restructuring of this educational resource. Some of the main features of this methodology - iterativeness, multiple participants, different validations and multidisciplinary collaboration - were absolutely essential for the final result, evidencing the relevance, suitability and effectiveness of the EDR approach for the development of these kinds of educational resources. Thus, the followed procedures and the obtained result seem to be aligned with the reference literature: EDR can lead to the development of high-quality DER, innovative and appropriate to contexts, proving to be a privileged methodology for the development of DER such as “Sustainability at Play” game.

Round 2

Reviewer 2 Report

The manuscript has included improvements that reflect the work and concern of the authors. Although there are aspects that could be improved, in the broad path of research, they can be taken as learning for future research. Forward!